# Pre-/perinatal reduced optimality and neurodevelopment at 1 month and 3 years of age: Results from the Japan Environment and Children's Study (JECS)

**Kahoko Yasumitsu-Lovell**[1,2], **Lucy Thompson**[1,3], **Elisabeth Fernell**[1], **Masamitsu Eitoku**[2], **Narufumi Suganuma**[2]*, **Christopher Gillberg**[1,2,3], **on behalf of the Japan Environment and Children's Study Group**¶

**1** Gillberg Neuropsychiatry Centre, Sahlgrenska Academy, University of Gothenburg, Gothenburg, Sweden, **2** Department of Environmental Medicine, Kochi Medical School, Kochi University, Kochi, Japan, **3** Institute of Applied Health Sciences, University of Aberdeen, Aberdeen, United Kingdom

¶ Membership of the Japan Environment and Children's Study Group is provided in the Acknowledgments.
* nsuganuma@kochi-u.ac.jp

**Data Availability Statement:** Data are unsuitable for public deposition due to ethical restrictions and

## Abstract

Neurodevelopmental disorders (NDDs) in children are associated with a complex combination of genetic and/or environmental factors. Pre-/perinatal events are major known environmental suboptimal factors, and their individual and combined contributions vary. This study investigated the association between pre-/perinatal reduced optimality and child development observed by parents at 1 month, as well as NDDs at 3 years of age (i.e., motor delay, intellectual disability, developmental language disorder, and autism spectrum disorder), in the context of the Japan Environment and Children's Study. The study also assessed whether child development at 1 month predicted NDDs at 3 years of age. Associations between 25 pre-/perinatal factors and (a) developmental concerns at 1 month of age and (b) NDDs at 3 years were analyzed ($n = 71,682$). Binomial regression models were used to investigate risk ratios of the developmental outcome at each time point for total pre-/perinatal reduced optimality scale scores, as well as for individual pre-/perinatal factors of the reduced optimality scale. Finally, we assessed the ability of parental observations of offspring development at 1 month to predict NDDs at 3 years. Total reduced optimality scores were positively associated with 1-month developmental concerns and 3-year NDDs, with higher scores (i.e., a reduction in optimality) associated with an increased risk of both NDDs and earlier parental concerns. Neonatal transportation, epidural analgesia, advanced maternal age, cesarean section delivery, Apgar score ≤8, and hyperbilirubinemia were identified as individual risk factors for 3-year NDDs, overlapping with 14 risk factors for 1-month developmental concerns except Apgar score ≤8. Among six developmental items assessed at 1 month of age, concerns about gross motor function and difficulty holding/trouble calming down had the strongest associations with later-diagnosed motor delay and autism spectrum disorder, respectively. Five perinatal factors and advanced maternal age were associated with NDD at 3 years of age, as were early parental developmental concerns regarding their offspring's overall development, indicating the importance of careful follow-up of offspring

legal framework of Japan. It is prohibited by the Act on the Protection of Personal Information (Act No. 57 of 30 May 2003, amendment on 9 September 2015) to publicly deposit the data containing personal information. Ethical Guidelines for Medical and Health Research Involving Human Subjects enforced by the Japan Ministry of Education, Culture, Sports, Science and Technology and the Ministry of Health, Labour and Welfare also restricts the open sharing of the epidemiologic data. All inquiries about access to data should be sent to Dr. Shoji F. Nakayama, JECS Programme Office, National Institute for Environmental Studies ((https://www.env.go.jp/chemi/ceh/en/index.html) via email (jecs-en@nies.go.jp).

**Funding:** The Japan Environment and Children's Study (JECS) is funded by the Ministry of the Environment (MOE) of Japan, including data collection. The present study utilized part of the JECS data. However, for this particular study "Pre-/perinatal reduced optimality and neurodevelopment at 1 month and 3 years of age – Results from the Japan Environment and Children's Study (JECS)", the funder had no role in study design, analysis, decision to publish, or preparation of the manuscript.

**Competing interests:** The authors have declared that no competing interests exist.

born with pre-/perinatal reduced optimality. The results also implicated early parental concerns, as early as 1 month, may also be a useful indicator of later NDD status.

## Introduction

The etiologies of neurodevelopmental disorders (NDDs) vary, with both genetic and environmental factors being involved. Among the environment factors, pre-/perinatal factors are the most crucial, as explained by the DOHaD (Developmental Origins of Health and Disease) hypothesis [1, 2]. To gain an understanding of the overall, and possibly additive, adverse effects during these crucial periods, some studies have used an obstetric optimality score, in which each pre-/perinatal factor is weighted equally [3–6]. The optimality concept was developed by Prechtl in 1968 to obtain information about the patterns of additive and interacting prejudicial events during the prepartum, partum, and postpartum periods [4, 7, 8]. This method overcomes the difficulties in grading the severity of a certain complication or in defining an abnormal situation. Other studies have examined individual risk factors separately to identify specific etiologic associations [1, 9–11]. Both approaches have advantages and disadvantages, and the results of previous studies of individual pre-/perinatal factors have been inconsistent.

The heterogeneity of the results from these previous studies is likely due to the considerable variations in study design, sample size, and diagnostic criteria, as well as to an insufficiency of information on confounding factors [12]. Large birth cohort studies overcome these challenges by prospectively collecting information on various suboptimal pre-/perinatal factors [13].

As treatments and interventions for neurodevelopmental and neuropsychiatric disorders have progressed, the need for early recognition, with a view to better outcomes, has grown [14, 15]. Hence, many studies have investigated early precursors of later-diagnosed NDDs [16]; however, to the best of our knowledge, no large birth cohort study has prospectively investigated the association between pre-/perinatal factors and offspring development assessed as early as 1 month of age, nor has any study explored whether parental developmental concerns at 1 month could indicate NDDs at 3 years of age [17].

The aim of this study was to investigate the associations between: (1) pre-/perinatal reduced optimality and parental developmental concerns at 1 month of age; (2) pre-/perinatal reduced optimality and NDDs at 3 years; and (3) developmental concerns at 1 month and NDDs at 3 years of age, by utilizing data from one of the world's largest ongoing national birth cohort studies, the Japan Environment and Children's Study (JECS).

## Methods

### Study design

This study was a longitudinal study within the JECS birth cohort. The main goal of the JECS is to investigate the association between environmental factors during the fetal and infancy periods and child health and development. Detailed information on the JECS has been published elsewhere [18]. In total, 104,062 pregnancies were registered in the JECS between January 2011 and March 2014, with 100,303 live births.

### Data collection

The study used the jecs-ta-20190930-qsn dataset released in October 2019, along with the supplementary dataset jecs-ta-20190930-qsn-add001. Data were collected from the time of

enrolment during pregnancy up until the child reached 1 month of age and included medical record transcriptions performed by physicians, midwives, and nurses (at study entry in the first trimester, at birth, and at the 1-month postnatal routine examination). In addition, parents completed self-administered questionnaires (twice during pregnancy, once 1 month after the birth, and biannual questionnaires from 6 months to 3 years of age). The medical record transcriptions contained pre-, peri-, and neonatal information, including maternal medical history, maternal age, pregnancy-related problems, parity, mode of delivery, and the newborn's well-being; the questionnaires covered various topics, including child development and diagnoses, parental lifestyle, socioeconomic status, and diet.

### Study participants

To ensure the representativeness of the JECS participants, 15 regional centers were selected from the northern (Hokkaido) to southern (Okinawa) ends of Japan. Participants were recruited at Co-operating healthcare providers or local government offices where pregnant women register themselves, with a targeted coverage rate of >50% in the Study Areas [18]. The baseline characteristics of the JECS mother–child dyads were comparable with those obtained in the 2013 national survey [19].

Of the 104,102 JECS enrollees, 71,682 were included in the study, following the criteria: participants with complete information on (1) pre-/perinatal factors constituting the optimality scale listed in Table 1; (2) development assessed by parents at 1 month of age: and (3) NDD diagnoses by 3 years of age (Fig 1).

### Exposure measurement

To assess the additive effects of non-optimal pre-/perinatal factors on neurodevelopment in the offspring at 1 month and 3 years of age, we used the concept of "reduced optimality," introduced by Prechtl in 1968 and thereafter developed into an instrument to detect intercorrelations between non-optimal factors and neonatal neurological abnormalities [7, 8]. Many researchers have recognized the usefulness of the concept in identifying potential neuropathogenic background factors affecting neurodevelopment in children and have used it in various studies [4, 20–22].

The "reduced optimality scale" that we used consisted of 25 items to accommodate the specific items available in the JECS data (Table 1). We defined "prenatal" as the period of pregnancy before the onset of labor and "perinatal" as the time from the onset of labor, including delivery, and the first 7 days of life. Items were given a score of 0 if they met the optimal conditions specified in Table 1 and a score of 1 if they fell outside the optimal range. Scores for individual items were summed to obtain the total score for reduced optimality during the pre-/perinatal periods. The higher total scores indicate the more *reduced* pre-/perinatal optimality.

### Outcome measurement

Information on child development at 1 month of age was based on six questions in the questionnaire answered by parents regarding their child's gross motor function, vision, hearing, crying, and reaction when being held. These items were selected based on previous findings that, from early life, individuals with NDDs frequently show a variety of often coexisting pathological or atypical clinical signs in the areas of communication, fine and gross motor function, and sensory reactions [13, 23]. These findings were conceptualized as the Early Symptomatic Syndromes Eliciting Neurodevelopmental Clinical Examinations (ESSENCE) by Gillberg in 2010 [24]. "ESSENCE" is an umbrella term that includes the whole picture of various coexisting difficulties affecting at least 10% of children aged <18 years [25].

**Table 1. Optimality scale: Pre- and perinatal factors and their optimal conditions.**

| Pre-/perinatal factor | Optimal condition |
|---|---|
| 1. Maternal age | 20–35 |
| 2. Parity | 1 or 2 |
| 3. History of spontaneous abortion | 0–2 |
| 4. Assisted reproductive technologies (ARTs)[a] | No |
| 5. Threatened abortion/threatened premature labor | Absent |
| 6. Antibiotic intake during pregnancy | Absent |
| 7. Pregnancy-induced hypertension (PIH) and hypertension | Absent |
| 8. Psychiatric specialist care | Absent |
| 9. Maternal disorders[b] | Absent |
| 10. Neuropsychotropic medication use | Absent |
| 11. Gestational age (weeks) | 37–40 |
| 12. Intrauterine growth restriction (IUGR) | No |
| 13. Small for gestational age (SGA) | No |
| 14. Twin or multiple birth | No |
| 15. Breech, foot, or other abnormal presentation | No |
| 16. Vacuum/forceps extraction | No |
| 17. Induced delivery | No |
| 18. Cesarean section delivery | No |
| 19. Epidural analgesia | No |
| 20. Length of labor (h) | 0–24 |
| 21. Apgar score (5 min) | 9 or 10 |
| 22. Umbilical cord/placental problems | No |
| 23. Meconium staining | No |
| 24. Neonatal transportation | No |
| 25. Hyperbilirubinemia | Absent |

[a]ARTs include ovulation induction, artificial insemination with the husband's sperm, in vitro fertilization, intracytoplasmic sperm injection, fresh embryo transfer, frozen embryo transfer, and blastocyst transfer.
[b]Maternal disorders include diabetes/gestational diabetes, epilepsy, and hyper- or hypothyroidism.

After scrutinizing the data, an experienced child psychiatrist (CG) and a child neurologist (EF) dichotomized the responses to the six questions when there were more than two options (0 = typical; 1 = concern) as follows: ability of baby to move his/her right and left limbs equally well ("gross motor function": yes = 0; no/uncertain = 1); reaction of baby to sound (e.g. parent's voice: "hearing"; yes = 0; no/uncertain = 1); apparent ability of baby to see things ("vision": yes = 0; no/uncertain = 1); frequency of difficulty holding the baby because of issues with his/her attachment or behavior, or both (e.g. crying, bending backwards; "difficulty holding": sometimes/seldom/never = 0; often = 1); intensity and frequency of crying ("intense/frequent crying": sometimes but short = 0; quite often and long, or hardly ever = 1); and trouble calming the crying baby ("trouble calming": no = 0; yes = 1). A maximum possible score is 6, indicating parental developmental concerns for all the six domains regarding the child's development. The total outcome score ("developmental concerns") was used as is (0–6), in addition to a dichotomized score using a cut-off of 1, with scores of 0 and 1 indicating "typical" development and scores ≥2 indicating "concern". The cut-off point for the dichotomization was based on the estimated prevalence of ESSENCE in children, which is believed to be in the range of 10%–20% [25–27]. In the present study, there was "concern" for 20.74% of children

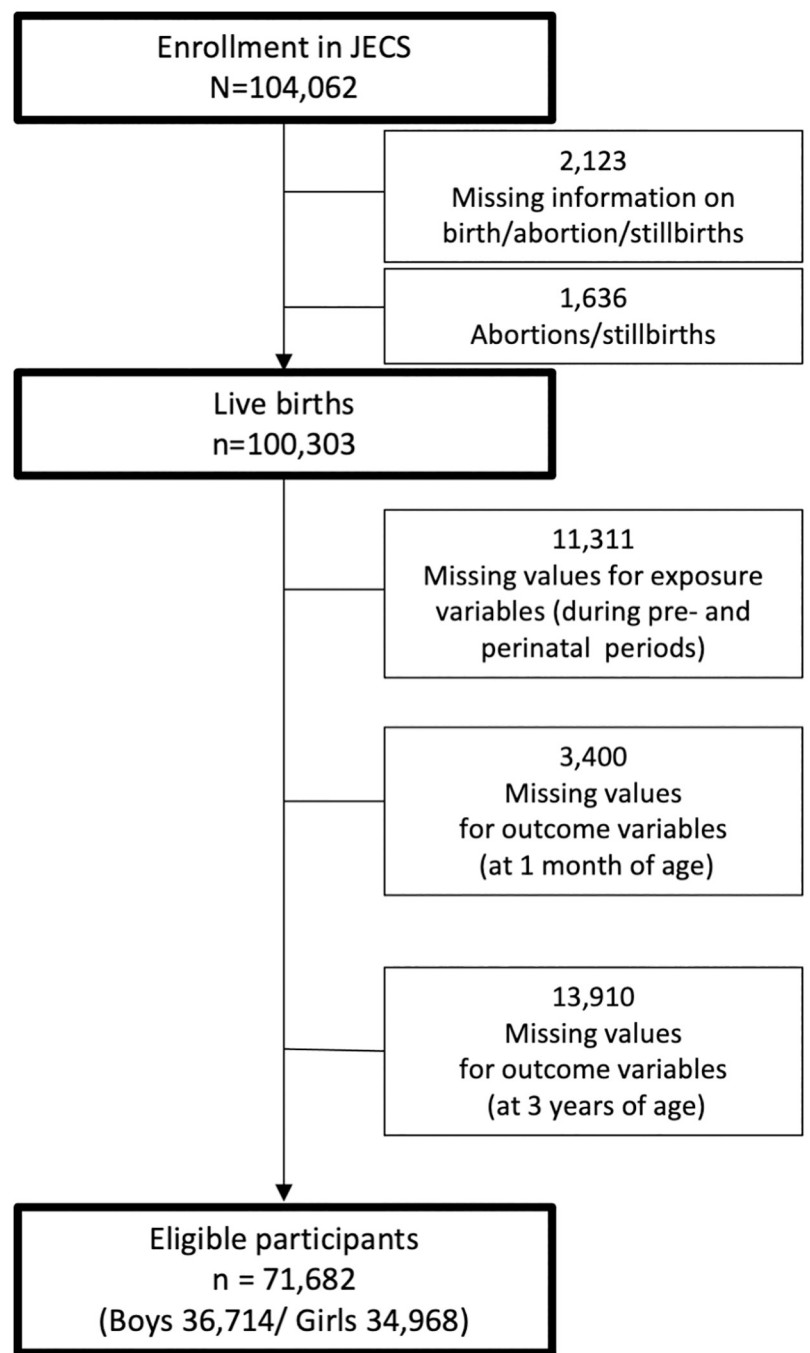

**Fig 1. Flow diagram showing enrolment of eligible children from among the Japan Environment and Children's Study (JECS) participants.**

(i.e., scores between 2 and 6; $n = 14,869$), compared with 79.26% of children considered "typical" (scores of 0 or 1; $n = 56,813$).

Information on NDD diagnoses at the age of 3 years was based on a question to parents about their 3-year-old child's medical and neurodevelopmental history in the previous year: "Has your child been diagnosed by doctors since turning 2 years old with any of the following

conditions? Please also include earlier diagnosis which has been followed-up." The list of potential conditions included cancer, immune-related problems such as allergies, infection, and NDDs. We defined NDDs as one or more parent-reported medical diagnoses of motor delay (MD); intellectual disability (ID) and/or developmental language disorder (DLD); and autism spectrum disorder (ASD).

## Statistical analysis

Associations between the reduced optimality scores and both developmental concerns at 1 month and NDDs at 3 years were examined by using binomial regression with a log-link function. We then separated epilepsy, diabetes, and thyroid disease, which were collapsed into a single item ("Maternal disorders") in the reduced optimality scale. Therefore, relative risks (RRs), with 95% confidence intervals (CIs), were calculated for 27 individual pre-/perinatal factors by using the log-binomial model. Socioeconomic factors (household income and maternal education) were added to the model to obtain adjusted RRs (aRRs) after having confirmed that there was no multicollinearity among explanatory variables in the model. As non-optimal condition of maternal age included both younger ($<20$ years old) and older ($>35$ years old) ends, and so did that of gestational age at birth ($<37$ weeks or $>41$ weeks), we recategorize these two items into three groups in sub-analysis, and calculated aRRs of these two factors. Multiple imputation by chained equations was conducted to confirm that the results of the final model, adjusted for family income and maternal education, were reliable, because 4,947 participants (7.0%) were not included in the final adjusted model.

We subsequently assessed whether the scores for development at 1 month, based on parental observations, predicted the presence of NDDs at 3 years of age by using receiver operating characteristic (ROC) curve analysis and calculating the area under the curve (AUC). Finally, the RRs of each of the six developmental concern items at 1 month for NDDs at 3 years were calculated by binomial regression with a log-link function. A two-tailed $p$ value of $< 0.05$ was considered statistically significant. All data analyses were performed by using Stata/MP version 16.0 (StataCorp LP, College Station, TX, USA).

## Ethics approval

The JECS protocol was reviewed and approved by the Ministry of the Environment's Institutional Review Board on Epidemiological Studies (No. 100910001) and the ethics committees of all participating institutions. Written informed consent was obtained from all participants.

## Results

Among the 71,682 study participants (36,714 boys, 34,968 girls), 750 children (1.05%) received at least one NDD diagnosis (MD; ID and/or DLD; and ASD) at 3 years of age, with boys ($n = 542$) outnumbering girls ($n = 208$). ID and/or DLD was diagnosed most often ($n = 487$), followed by ASD ($n = 329$) and MD ($n = 172$). Of the children diagnosed with NDDs, 530 were diagnosed with one NDD, 202 were diagnosed with two NDDs, and 18 were diagnosed with three NDDs. Diagnoses of multiple NDDs always included ID and/or DLD (Table 2). The NDD group had a significantly higher proportion of pre-/perinatal reduced optimality than the non-NDD group for 16 scale items (Table 3).

**Table 2. Prevalence of neurodevelopmental disorders in children (*n* = 71,682).**

| NDD diagnosis | No. (*n* = 750, 1.05%) with | | | No. (%) with each diagnosis |
|---|---|---|---|---|
| | 1 NDD | 2 NDDs | 3 NDDs | |
| **MD** | 32 | | | 172 (0.24) |
| **ID and/or DLD** | 267 | | | 487 (0.68) |
| **ASD** | 231 | | | 329 (0.46) |
| **MD + ID and/or DLD** | | 122 | | |
| **MD + ASD** | | 0 | | |
| **ID and/or DLD + ASD** | | 80 | | |
| **MD + ID and/or DLD + ASD** | | | 18 | |
| **Total** | 530 | 202 | 18 | |

ASD, autism spectrum disorder; DLD, developmental language disorder; ID, intellectual disability; MD, motor delay.

**Table 3. Prevalence of pre- or perinatal factors in participants with and without a neurodevelopmental disorder diagnosis.**

| Suboptimal pre- or perinatal factor | Optimal condition | No. with non-optimal score (*n* = 71,682) | No. (%) with NDD diagnosis (*n* = 750) | No. (%) without NDD diagnosis (*n* = 70,932) | *p*-value |
|---|---|---|---|---|---|
| **1. Maternal age** | 20–35 | 16,284 | 239 (31.87) | 16,045 (22.62) | <0.001 |
| **2. Parity** | 1 or 2 | 32,604 | 382 (50.93) | 32,222 (45.43) | 0.003 |
| **3. History of spontaneous abortion** | 0–2 | 613 | 8 (1.06) | 605 (0.85) | 0.527 |
| **4. ARTs** | No | 5259 | 80 (10.67) | 5,179 (7.30) | <0.001 |
| **5. Threatened abortion/ premature labor** | Absent | 19,453 | 232 (30.93) | 19,221 (27.10) | 0.019 |
| **6. Antibiotic during pregnancy** | Absent | 15,829 | 169 (22.53) | 15,660 (22.08) | 0.765 |
| **7. PIH and hypertension** | Absent | 2,555 | 42 (5.59) | 2,513 (3.54) | 0.003 |
| **8. Psychiatric problems** | No | 538 | 12 (1.60) | 526 (0.74) | 0.007 |
| **9–1. Diabetes/GDM** | Absent | 2,313 | 28 (3.72) | 2,285 (3.22) | 0.430 |
| **9–2. Epilepsy** | Absent | 178 | 3 (0.40) | 175 (0.25) | 0.401 |
| **9–3. Thyroidism (hyper-/hypo-)** | Absent | 1020 | 18 (2.39) | 1002 (1.41) | 0.023 |
| **10. Neuropsychotropic medication use** | No | 781 | 14 (1.86) | 767 (1.08) | 0.039 |
| **11. Gestational age** | 37–40 | 10,065 | 156 (20.74) | 9,909 (13.97) | <0.001 |
| **12. IUGR** | No | 1,428 | 32 (4.27) | 1,396 (1.97) | <0.001 |
| **13. SGA** | No | 8,678 | 153 (20.35) | 8,525 (12.02) | <0.001 |
| **14. Twins or multiple birth** | No | 1,221 | 19 (2.53) | 1,202 (1.69) | 0.077 |
| **15. Abnormal fetal presentations** | No | 2,881 | 40 (5.33) | 2,841 (4.01) | 0.065 |
| **16. Vacuum/forceps extraction** | No | 4,228 | 48 (6.38) | 4,180 (5.89) | 0.558 |
| **17. Induced delivery** | No | 12,500 | 140 (18.67) | 12,360 (17.44) | 0.373 |
| **18. Cesarean section delivery** | No | 13,608 | 207 (27.60) | 13,401 (18.89) | <0.001 |
| **19. Epidural analgesia** | No | 1,530 | 24 (3.19) | 1,506 (2.12) | 0.042 |
| **20. Labor >24 h** | <24 | 2,337 | 22 (2.93) | 2,315 (3.26) | 0.612 |
| **21. Apgar score (5 min)** | 9 or 10 | 3,727 | 82 (10.93) | 3,645 (5.14) | <0.001 |
| **22. Umbilical cord/placental problems** | No | 16,924 | 175 (23.33) | 16,749 (23.61) | 0.858 |
| **23. Meconium staining** | No | 2,447 | 34 (4.53) | 2,413 (3.40) | 0.090 |
| **24. Neonatal transportation** | No | 4,064 | 124 (16.53) | 3,940 (5.55) | <0.001 |
| **25. Hyperbilirubinemia** | Absent | 7,899 | 133 (17.73) | 7,766 (10.95) | <0.001 |

ARTs, assisted reproductive technologies; GDM, gestational diabetes; IUGR, intrauterine growth restriction; NDD, neurodevelopmental disorder; PIH, pregnancy-induced hypertension; SGA, small for gestational age.

**Table 4. Reduced optimality scale scores versus outcomes as assessed by parents at 1 month and versus diagnoses of neurodevelopmental disorders at 3 years of age.**

| Reduced optimality Scale Scores | No. of participants | At 1 month of age[a] | | | | At 3 years of age | | | |
|---|---|---|---|---|---|---|---|---|---|
| | | "Concerns" | (%) | "Typical" | (%) | NDDs | (%) | No NDDs | (%) |
| 0 | 6,930 | 825 | (11.90) | 6,105 | (88.10) | 33 | (0.48) | 6,897 | (99.52) |
| 1 | 14,773 | 2,324 | (15.73) | 12,449 | (84.27) | 111 | (0.75) | 14,662 | (99.25) |
| 2 | 16,340 | 3,211 | (19.65) | 13,129 | (80.35) | 135 | (0.83) | 16,205 | (99.17) |
| 3 | 13,505 | 3,109 | (23.02) | 10,396 | (76.98) | 149 | (1.10) | 13,356 | (98.90) |
| 4 | 9,054 | 2,219 | (24.51) | 6,835 | (75.49) | 114 | (1.26) | 8,940 | (98.74) |
| 5 | 5,329 | 1,488 | (27.92) | 3,841 | (72.08) | 82 | (1.54) | 5,247 | (98.46) |
| 6 | 2,857 | 801 | (28.04) | 2,056 | (71.96) | 55 | (1.93) | 2,802 | (98.07) |
| 7 | 1,435 | 421 | (29.34) | 1,014 | (70.66) | 26 | (1.81) | 1,409 | (98.19) |
| ≥8 | 1,459 | 471 | (32.28) | 988 | (67.72) | 45 | (3.08) | 1,414 | (96.92) |
| Total | 71,682 | 14,869 | (20.74) | 56,813 | (79.26) | 750 | (1.05) | 70,932 | (98.95) |
| Mean[b] | | 3.11 | | 2.55 | | 3.48 | | 2.65 | |

[a]Information on child development at 1 month of age was based on six questions. A maximum possible score of 6 indicated several concerns regarding the child's development. Scores were dichotomized by using a cut-off of 1, with scores of 0 and 1 indicating "typical" development and scores ≥2 indicating "concern".
[b]The mean scores were significantly different between "Concerns" and "Typical" as well as NDDs and no NDDs ($p<0.001$).
NDDs, neurodevelopmental disorders

## Reduced optimality scale scores and developmental concerns at 1 month/ NDDs at 3 years of age

The reduced optimality scale scores were strongly skewed to the right, with a median of 2 (Table 4). Mean reduced optimality scale scores differed significantly between children in the "concern" and "typical" groups at 1 month of age (3.11 vs. 2.55, respectively; $p < 0.001$), as well as with and without NDDs at 3 years of age (3.48 vs. 2.65, respectively; $p < 0.001$).

RRs increased significantly with increasing scores on the reduced optimality scale for both developmental concerns at 1 month and NDDs at 3 years (Table 5). The discrepancy between the lowest and highest RRs for NDDs at 3 years of age was much wider than that for development at 1 month. When a reduced optimality score of 0 was used as the reference, for developmental concerns at 1 month of age, the RRs were 1.32 (95% CI 1.23–1.42) for a score of

**Table 5. Relative risks of developmental concerns at 1 month of age and of a diagnosis of neurodevelopmental disorder at 3 years of age for each suboptimality scale score.**

| Reduced optimality scale score | Developmental concerns at 1 month of age | | NDD at 3 years of age | |
|---|---|---|---|---|
| | RR (95% CI) | *p*-value | RR (95% CI) | *p*-value |
| 0 | Reference | | Reference | |
| 1 | 1.32 (1.23–1.42) | <0.001 | 1.58 (1.07–2.32) | 0.021 |
| 2 | 1.65 (1.54–1.77) | <0.001 | 1.74 (1.19–2.54) | 0.004 |
| 3 | 1.93 (1.80–2.08) | <0.001 | 2.32 (1.59–3.37) | <0.001 |
| 4 | 2.06 (1.91–2.22) | <0.001 | 2.64 (1.80–3.89) | <0.001 |
| 5 | 2.35 (2.17–2.53) | <0.001 | 3.23 (2.16–4.83) | <0.001 |
| 6 | 2.36 (2.16–2.57) | <0.001 | 4.04 (2.63–6.21) | <0.001 |
| 7 | 2.46 (2.22–2.73) | <0.001 | 3.80 (2.28–6.34) | <0.001 |
| ≥8 | 2.71 (2.46–2.99) | <0.001 | 6.48 (4.15–10.11) | <0.001 |

CI, confidence interval; NDD, neurodevelopmental disorder; RR, relative risk.

1 and 2.71 (95% CI 2.46–2.99) for a score ≥8,and the RRs of NDDs at 3 years of age were 1.58 (95% CI 1.07–2.32) for a score of 1 and 6.48 (95% CI 4.15–10.11) for a score ≥8; in comparison.

## Association between each of the pre-/perinatal factors of the reduced optimality scale and developmental concerns at 1 month/NDDs at 3 years of age

When the aRRs of developmental concerns at 1 month for each pre-/perinatal item on the reduced optimality scale were examined in generalized linear models, there were 14 statistically significant pre-/perinatal factors with an aRR >1.00 (listed in order of decreasing aRR): nulliparity/high parity, epilepsy, neuropsychotropic medication, artificial reproductive technologies, epidural analgesia, twin/multiple birth, meconium staining, vacuum/forceps extraction, cesarean section delivery, small for gestational age, advanced maternal age, neonatal transportation, hyperbilirubinemia, and threatened abortion/premature labor (S1 Table).

For NDDs at 3 years of age, six items showed statistically significant associations (listed from highest to lowest aRR): neonatal transportation, epidural analgesia, young/advanced maternal age, cesarean section delivery, Apgar score ≤ 8, and hyperbilirubinemia (Table 6). Of the six pre-/perinatal reduced optimal factors for NDDs at 3 years of age, Apgar score was the only one that was not a risk factor for developmental concerns at 1 month of age; the remaining reduced optimal factors at 3 years of age were already risk factors for developmental concerns at 1 month (S1 Table).

The aRRs of a diagnosis of NDDs for these six risk factors showed different trends when each of the three different NDD diagnoses (i.e., MD, ID and/or DLD, and ASD) was examined separately. Whereas young/advanced maternal age and cesarean section delivery had the third- and fourth-highest aRRs of total NDD diagnoses and were also risk factors for each of the three different NDD diagnoses separately, epidural analgesia and nulliparity/high parity were risk factors only for ASD. Neonatal transportation, small for gestational age, and Apgar score ≤8 were risk factors for both MD and ID and/or DLD. Maternal hyper-/hypothyroidism was a risk factor only for MD. Hyperbilirubinemia was a risk factor only when all three NDDs were combined. Because the optimal maternal age range was defined as 20 to 35 years inclusive (n = 55,398; 77.28%) and the non-optimal maternal age included both age <20 years (n = 403; 0.56%) and age >35 years (n = 15,881; 22.5%) (Table 3), maternal age was recategorized into three groups: <20, ≥20 to ≤35, and >35 years. Only the oldest group, and not the youngest group, had an aRR >1.00 (RR 1.50; 95% CI 1.27–1.77; p < 0.001). Similar to the definition of non-optimal maternal age (<20 or >35 years old), non-optimal gestational age at birth included both pre-term (<37 weeks, n = 3,448, 4.9% of the whole participants) and post-term (≥41 weeks n = 6,577, 9.2%). To investigate possible different risk of pre- and post-term birth, gestational age was recategorized into three groups: <37 weeks (pre-term), ≥37to <41 weeks (term), and ≥41 weeks (post-term). The proportion of children with NDDs in each of the three groups were 2.1%, 1.0%, and 1.3% respectively. In the bivariate analysis, crude RRs for the pre- and post-term groups for 3-year NDDs were 2.20 (95%CI 1.73–2.79, p<0.001), and 1.30 (95%CI 1.03 1.64, p<0.05) respectively. However, neither aRRs of the pre- and post-term groups in the final analysis showed statistically significant association (pre-term: aRR 0.81, 95%CI 0.56–1.17, p = 0.259; post-term: aRR 1.21, 95%CI 0.95–1.56, p = 0.113).

## Association between developmental concerns at 1 month and NDDs at 3 years of age

When associations between "concern" scores in each of the six developmental items (gross motor function, hearing, vision, difficulty holding, intense/frequent crying, and trouble

**Table 6. Adjusted relative risk of a diagnosis of neurodevelopmental disorder at 3 years of age for each pre-/perinatal factor.**

| | NDD diagnosis at 3 years of age (n = 750) | | MD (n = 172) | | ID and/or DLD (n = 487) | | ASD (n = 329) | |
|---|---|---|---|---|---|---|---|---|
| | aRR (95% CI) | p-value | aRR (95% CI) | p-value | aRR (95% CI) | p-value | aRR (95% CI) | p-value |
| 1. Young/Advanced maternal age | 1.48 (1.26–1.75) | <0.001 | 1.46 (1.04–2.04) | 0.028 | 1.53 (1.25–1.87) | <0.001 | 1.33 (1.03–1.72) | 0.029 |
| 2. Nulliparity/high parity | 1.16 (0.99–1.35) | 0.064 | 1.08 (0.78–1.50) | 0.627 | 0.96 (0.79–1.17) | 0.718 | 1.51 (1.19–1.92) | 0.001 |
| 3. History of spontaneous abortion | 1.10 (0.55–2.21) | 0.782 | 1.58 (0.50–4.92) | 0.434 | 0.60 (0.19–1.88) | 0.384 | 0.70 (0.17–2.82) | 0.619 |
| 4. ARTs | 1.13 (0.88–1.45) | 0.357 | 1.09 (0.66–1.77) | 0.745 | 1.18 (0.87–1.61) | 0.287 | 1.03 (0.69–1.54) | 0.870 |
| 5. Threatened abortion/premature labor | 1.09 (0.92–1.28) | 0.321 | 1.14 (0.81–1.59) | 0.448 | 1.13 (0.93–1.39) | 0.222 | 0.93 (0.72–1.21) | 0.587 |
| 6. Antibiotic during pregnancy | 0.98 (0.82–1.18) | 0.855 | 1.13 (0.79–1.60) | 0.512 | 1.05 (0.85–1.31) | 0.645 | 0.93 (0.70–1.22) | 0.586 |
| 7. PIH and hypertension | 1.00 (0.72–1.40) | 0.984 | 0.71 (0.36–1.42) | 0.332 | 1.22 (0.83–1.78) | 0.318 | 0.89 (0.50–1.56) | 0.673 |
| 8. Psychiatric problems | 1.76 (0.92–3.36) | 0.085 | 1.05 (0.28–3.92) | 0.940 | 1.84 (0.86–3.93) | 0.114 | 1.41 (0.46–4.27) | 0.547 |
| 9–1. Diabetes/GDM | 0.87 (0.58–1.30) | 0.495 | 0.59 (0.24–1.45) | 0.252 | 0.82 (0.50–1.35) | 0.440 | 1.56 (0.95–2.55) | 0.078 |
| 9–2. Epilepsy | 1.30 (0.39–4.31) | 0.671 | 0.87 (0.11–7.21) | 0.900 | 1.70 (0.49–5.82) | 0.401 | 1.07 (0.13–8.56) | 0.951 |
| 9–3. Thyroidism (hyper-/hypo-) | 1.34 (0.82–2.20) | 0.241 | 2.15 (1.01–4.57) | 0.046 | 1.28 (0.68–2.38) | 0.442 | 1.05 (0.43–2.54) | 0.914 |
| 10. Neuropsychotropic medication use | 1.18 (0.63–2.24) | 0.602 | 2.39 (0.88–6.51) | 0.089 | 1.46 (0.72–2.98) | 0.299 | 1.14 (0.40–3.25) | 0.803 |
| 11. Gestational age | 1.08 (0.87–1.34) | 0.504 | 0.84 (0.54–1.31) | 0.439 | 0.95 (0.72–1.25) | 0.695 | 1.11 (0.79–1.55) | 0.552 |
| 12. IUGR | 1.33 (0.89–1.98) | 0.165 | 1.82 (0.99–3.36) | 0.055 | 1.41 (0.89–2.24) | 0.138 | 0.71 (0.29–1.75) | 0.454 |
| 13. SGA | 1.17 (0.92–1.48) | 0.198 | 1.76 (1.12–2.75) | 0.014 | 1.46 (1.10–1.94) | 0.008 | 0.97 (0.66–1.44) | 0.888 |
| 14. Twins or multiple birth | 0.73 (0.44–1.20) | 0.214 | 0.36 (0.13–1.01) | 0.052 | 0.66 (0.36–1.22) | 0.187 | 0.84 (0.36–1.97) | 0.685 |
| 15. Abnormal fetal presentations | 0.82 (0.57–1.18) | 0.290 | 0.67 (0.33–1.36) | 0.269 | 0.82 (0.52–1.28) | 0.377 | 0.86 (0.49–1.50) | 0.595 |
| 16. Vacuum/forceps extraction | 1.19 (0.86–1.63) | 0.290 | 1.65 (0.88–3.10) | 0.120 | 1.12 (0.74–1.68) | 0.590 | 1.09 (0.67–1.75) | 0.732 |
| 17. Induced delivery | 1.19 (0.97–1.47) | 0.103 | 1.37 (0.87–2.16) | 0.169 | 1.07 (0.82–1.40) | 0.626 | 1.15 (0.84–1.58) | 0.369 |
| 18. Cesarean section delivery | 1.41 (1.16–1.73) | 0.001 | 1.69 (1.14–2.53) | 0.010 | 1.35 (1.05–1.72) | 0.018 | 1.56 (1.15–2.11) | 0.004 |
| 19. Epidural analgesia | 1.55 (1.03–2.33) | 0.037 | 1.66 (0.73–3.78) | 0.230 | 1.33 (0.76–2.31) | 0.320 | 1.99 (1.15–3.42) | 0.013 |
| 20. Labor >24 h | 0.89 (0.57–1.38) | 0.605 | 0.57 (0.18–1.82) | 0.344 | 0.89 (0.49–1.59) | 0.683 | 0.95 (0.51–1.75) | 0.857 |
| 21. Apgar Score (5 min) | 1.38 (1.07–1.79) | 0.014 | 1.86 (1.22–2.85) | 0.004 | 1.46 (1.07–1.99) | 0.016 | 1.06 (0.66–1.69) | 0.823 |
| 22. Umbilical cord/placenta problems | 0.91 (0.76–1.08) | 0.279 | 0.86 (0.60–1.22) | 0.394 | 0.83 (0.67–1.04) | 0.108 | 0.98 (0.75–1.28) | 0.874 |
| 23. Meconium staining | 1.21 (0.85–1.74) | 0.288 | 1.62 (0.86–3.02) | 0.132 | 1.38 (0.89–2.13) | 0.149 | 1.13 (0.64–1.98) | 0.674 |
| 24. Neonatal transportation | 2.30 (1.81–2.92) | <0.001 | 6.32 (4.25–9.40) | <0.001 | 2.39 (1.79–3.20) | 0.000 | 1.19 (0.75–1.88) | 0.454 |
| 25. Hyperbilirubinemia | 1.24 (1.01–1.53) | 0.040 | 0.95 (0.63–1.44) | 0.809 | 1.26 (0.97–1.63) | 0.078 | 1.31 (0.95–1.81) | 0.104 |

aRR, adjusted relative risk (adjusted for family income and maternal education); ARTs, assisted reproductive technologies; ASD, autism spectrum disorder; CI, confidence interval; DLD, developmental language disorder; GDM, gestational diabetes; ID, intellectual disability; IUGR, intrauterine growth restriction; MD, motor delay; NDD, neurodevelopmental disorder; PIH, pregnancy-induced hypertension; SGA, small for gestational age.

calming) at 1 month of age and each NDD at 3 years of age were examined, some differences were identified in MD and ASD. There was a significant likelihood that children with MD at 3 years of age had already been observed to have gross motor function problems at 1 month of age (RR 2.43; 95% CI 1.52–3.86; $P < 0.001$) (Table 7). Five of the six items observed at 1 month (excluding gross motor function delay) had statistically significant RRs >1.00 among children who were later diagnosed with ASD, with the "difficulty holding" item having the highest RR (2.08; 95% CI 1.48–2.91; $P < 0.001$), followed by "trouble calming," "hearing," "intense/frequent crying," and "vision" (Table 7). Even though the RR of any NDD at age 3 for the total score of developmental concern scale at 1 month was 1.56 (95% CI 1.33–1.82; $P < 0.001$) and the RRs for all six individual developmental concern items on the outcome scale were >1 with $P < 0.05$ (Table 7), the ability of the outcome scale scores at 1 month to predict NDDs at 3 years of age was not high, with an area under the receiver operating characteristic curve of 0.5593.

**Table 7. Relative risk of a diagnosis of neurodevelopmental disorder at 3 years of age for each "developmental concern" item observed at 1 month of age.**

|  | Any NDD diagnosis | | MD | | ID and/or DLD | | ASD | |
|---|---|---|---|---|---|---|---|---|
|  | RR 95% CI | *p*-value | RR 95% CI | *p*-value | RR 95% CI | *p*-value | RR 95% CI | *p*-value |
| **1. Gross motor function** | 1.63 (1.26–2.12) | <0.001 | 2.43 (1.52–3.86) | <0.001 | 1.69 (1.23–2.33) | 0.001 | 1.13 (0.71–1.79) | 0.604 |
| **2. Hearing** | 1.76 (1.43–2.17) | <0.001 | 1.95 (1.28–2.97) | 0.002 | 1.80 (1.39–2.32) | <0.001 | 1.72 (1.25–2.37) | 0.001 |
| **3. Vision** | 1.29 (1.10–1.52) | 0.002 | 1.40 (1.00–1.96) | 0.050 | 1.28 (1.05–1.57) | 0.017 | 1.38 (1.08–1.76) | 0.010 |
| **4. Difficulty holding** | 1.48 (1.15–1.91) | 0.001 | 0.98 (0.52–1.86) | 0.956 | 1.16 (0.81–1.64) | 0.418 | 2.08 (1.48–2.91) | <0.001 |
| **5. Intense/frequent crying** | 1.42 (1.21–1.67) | <0.001 | 1.59 (1.15–2.21) | 0.005 | 1.35 (1.11–1.65) | 0.003 | 1.45 (1.14–1.84) | 0.002 |
| **6. Trouble calming** | 1.36 (1.15–1.60) | <0.001 | 1.48 (1.05–2.08) | 0.025 | 1.03 (0.82–1.29) | 0.812 | 1.77 (1.40–2.24) | <0.001 |
| **Total outcome scores** | 1.56 (1.33–1.82) | <0.001 | 1.89 (1.38–2.60) | <0.001 | 1.42 (1.16–1.73) | 0.001 | 1.62 (1.28–2.05) | <0.001 |

ASD, autism spectrum disorder; DLD, developmental language disorder; ID, intellectual disabilities; MD, motor delay; NDD, neurodevelopmental disorder; RR, relative risk.

## Discussion

To our knowledge, this is the first large-scale study to have investigated the associations between pre-/perinatal factors and offspring development assessed by parents as early as 1 month of age, and to have explored whether parental observations at 1 month could indicate an NDD diagnosis at 3 years of age. A variation of the optimality scale, which has long been used by many researchers, and a range of prospectively collected information enabled us to comprehensively investigate the associations between reduced pre-/perinatal optimality and both development at 1 month and NDDs at 3 years of age. We found that: (1) pre-/perinatal reduced optimality dose-dependently affected children's neurodevelopment; (2) young/ advanced maternal age and cesarean section delivery were shared risk factors among the different NDD diagnoses, whereas other factors were characteristic to individual NDDs (e.g. epidural analgesia and nulliparity/high parity were risk factors only for ASD; maternal hyper-/ hypothyroidism was a risk factor only for MD; and neonatal care, Apgar score ≤8, and small for gestational age were shared risk factors between MD and ID and/or DLD); and (3) parents seemed able to perceive some signs of later-diagnosed NDDs in their children as early as 1 month of age.

A considerable number of previous studies have examined the association between pre-/ perinatal reduced optimality and NDDs, such as ASD, attention-deficit/hyperactivity disorder, ID, and cerebral palsy [3, 9, 10, 28, 29]. Similar associations were found here, particularly in the case of perinatal factors such as neonatal transportation, epidural analgesia, cesarean section delivery, Apgar scores ≤8, and hyperbilirubinemia. Of the six factors found to be associated with NDDs at 3 years of age in this study, only one—advanced maternal age—was prenatal. Even though causality cannot be established (e.g., the fetus may have already developed neurodevelopmental problems that were only diagnosed later, when they surfaced as "perinatal problems," or the perinatal problems may have adversely affected the offspring's brain), the results point towards the importance of medical care, particularly during the perinatal period, as well as the need to follow up children who survive adversities at birth.

Advanced maternal age (>35 years) and cesarean section delivery were the two risk factors shared among all three NDDs (MD, ID and/or DLD, and ASD). The effects of maternal age on offspring have been studied extensively from various perspectives, including social and biological [28–37], and advanced maternal age is known to be a detrimental biological factor owing to the rapid decline in healthy oocytes in women aged >35 years and the greater prevalence of pre-/perinatal complications with aging. Furthermore, maternal and paternal ages are highly

correlated, and advanced paternal age itself has also been perceived as a risk factor in many studies [35]. Parental young age is also a possible risk for offspring's NDDs, such as ADHD [38]. Maternal younger age was not a significant risk in our study, most likely due to the small number of the mothers younger than 20 years of age, and/or insufficient data on ADHD diagnosis at age 3 years. Regarding the second shared risk factor, namely cesarean section delivery, it may not be the cesarean section itself, but rather the reasons behind the need for this mode of delivery (e.g. emergency cesarean section delivery due to fetal distress and stalled labor) that pose a more fundamental risk of NDDs in offspring [39, 40].

Although the aRR for epidural analgesia was the second highest for total NDDs and the highest for ASD, these results must be interpreted carefully, and further investigations are required. Recent studies have investigated the risk of ASD posed by epidural analgesia, particularly after Qiu et al. reported that epidural analgesia increased the risk of ASD by 37%, on the basis of their study of 147,895 participants in northern California [41]. However, subsequent Canadian ($n$ = 123,175) and Danish ($n$ = 479,178) cohort studies concluded that epidural analgesia administered during labor was not associated with an increased risk of ASD in the offspring [42, 43], and other researchers have questioned the statistical methods used by Qiu et al.—particularly that they did not sufficiently account for residual confounding factors [43–46]. In Japan, epidural analgesia is not regularly used during labor and delivery, as evidenced by the rates in the present study (2.13%). The use of epidural analgesia could be an indication that women have other risks, and therefore require epidural analgesia. For example, they could have an extreme fear of pain or delivery because of their own neurodevelopmental/psychiatric problems, a tendency that was also found in a Danish cohort study [43]. In addition, medical professionals may choose to administer epidural analgesia because of perinatal complications, which are major risk factors for NDDs.

In our study, even though the total scores of the developmental concern scale at 1 month, which was neither validated nor conducted by specialists, did not accurately predict NDD diagnoses at 3 years of age, each of the six items in the scale may have indicated some neurodevelopmental problem as early as 1 month of age. Our findings suggest that parental observations could be a useful source of information for the early detection of, and intervention in, neurodevelopmental problems. People with ASD are known to have sensory issues (hyper-/hyposensitivity) and problems regulating emotions, which may explain the high RR among children with ASD for all but the gross motor function item at 1 month of age.

The strengths of this study lie in its large sample size with prospectively collected information from the prenatal period, which enabled us to include a range of information. However, four major limitations need to be noted. First, exposure and outcome information, or both, on developmental concerns was missing for 28,466 of 100,303 live births. Nevertheless, the mean maternal age at delivery—one of the major characteristics affecting developmental outcomes —in the groups included and excluded in this study was 31.2 and 30.8 years, respectively, suggesting that it is safe to assume that the two groups were similar and that the study participants were representative of the JECS participants as a whole. Second, the neurodevelopment of children at 1 month of age was assessed based on parental observations and was not corroborated by clinicians, and it did not include observations of other possible signs of developmental concerns, such as feeding and sleeping issues. Nonetheless, parental observations have been used in many studies and public health settings, and the developmental concern items in this study seem to be practically useful for providing outcome data at this early age. Third, the information regarding NDD diagnoses was based on the parental questionnaire and was not obtained from a medical database. The reported NDDs are most likely accurate and unlikely to be over-reported, because the question regarding the child's medical history specifically asked about "diagnoses by medical doctors." However, one cannot exclude a possibility of underreporting

due to the stigma and secrecy coping followed by any NDD diagnosis at this early stage of life [47, 48]. Fourth, even though the pre-/perinatal periods are critical windows of neurodevelopment, not all the risk factors were incorporated in the analysis, including those during the first years of child life prior to the NDD diagnosis.

## Conclusions

Our findings indicate that pre-/perinatal reduced optimality is likely associated, in a dose-dependent manner, with NDDs diagnosed in children at 3 years of age and that risk factors, except advanced maternal age and cesarean section delivery, vary for different NDD diagnoses. In addition, parental developmental concerns as early as 1 month after birth may be able to predict later diagnoses of NDDs. Because the children in this study were only 3 years old and relatively mild cases of NDDs may be more likely to be identified as the children get older, further follow-up of this cohort is crucial to confirm our findings.

## Supporting information

**S1 Table. Adjusted relative risks of developmental concerns at 1 month for each pre/perinatal factor.** aRR, adjusted relative risk (adjusted for family income and maternal education); ARTs, assisted reproductive technologies; CI, confidence interval; GDM, gestational diabetes; IUGR, intrauterine growth restriction; NDD, neurodevelopmental disorder; PIH, pregnancy-induced hypertension; SGA, small for gestational age.
(DOCX)

## Acknowledgments

The authors thank the JECS participants and the JECS Group as well as Nagamasa Maeda, Mikiya Fujieda, Naomi Mitsuda, and Atsuko Mori of the Kochi Regional Centre of the JECS and Sifa Marie Joelle Muchanga of the National Center for Global Health and Medicine. The members of the JECS Group as of 2022 are as follows: Michihiro Kamijima (principal investigator, Nagoya City University, Nagoya, Japan, jecs-en@nies.go.jp), Shin Yamazaki (National Institute for Environmental Studies, Tsukuba, Japan), Yukihiro Ohya (National Center for Child Health and Development, Tokyo, Japan), Reiko Kishi (Hokkaido University, Sapporo, Japan), Nobuo Yaegashi (Tohoku University, Sendai, Japan), Koichi Hashimoto (Fukushima Medical University, Fukushima, Japan), Chisato Mori (Chiba University, Chiba, Japan), Shuichi Ito (Yokohama City University, Yokohama, Japan), Zentaro Yamagata (Universit y of Yamanashi, Chuo, Japan), Hidekuni Inadera (University of Toyama, Toyama, Japan), Takeo Nakayama (Kyoto University, Kyoto, Japan), Tomotaka Sobue (Osaka University, Suita, Japan), Masayuki Shima (Hyogo Medical University, Nishinomiya, Japan), Hiroshige Nakamura(Tottori University, Yonago, Japan), Narufumi Suganuma (Kochi University, Nankoku, Japan), Koichi Kusuhara (University of Occupational and Environmental Health, Kitakyushu, Japan), and Takahiko Katoh (Kumamoto University, Kumamoto, Japan).

## Author Contributions

**Conceptualization:** Kahoko Yasumitsu-Lovell, Elisabeth Fernell, Christopher Gillberg.

**Data curation:** Kahoko Yasumitsu-Lovell, Masamitsu Eitoku.

**Formal analysis:** Kahoko Yasumitsu-Lovell, Masamitsu Eitoku.

**Funding acquisition:** Narufurmi Suganuma.

**Investigation:** Kahoko Yasumitsu-Lovell, Lucy Thompson, Narufurmi Suganuma.

**Methodology:** Kahoko Yasumitsu-Lovell, Elisabeth Fernell, Narufurmi Suganuma, Christopher Gillberg.

**Project administration:** Kahoko Yasumitsu-Lovell, Masamitsu Eitoku.

**Supervision:** Lucy Thompson, Elisabeth Fernell, Masamitsu Eitoku, Narufurmi Suganuma, Christopher Gillberg.

**Validation:** Kahoko Yasumitsu-Lovell, Elisabeth Fernell, Narufurmi Suganuma, Christopher Gillberg.

**Visualization:** Kahoko Yasumitsu-Lovell.

**Writing – original draft:** Kahoko Yasumitsu-Lovell.

**Writing – review & editing:** Kahoko Yasumitsu-Lovell, Lucy Thompson, Elisabeth Fernell, Masamitsu Eitoku, Narufurmi Suganuma, Christopher Gillberg.

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
