## [Decision Letter · Decision Letter 0]

23 Nov 2022

PONE-D-22-26962Pre-/perinatal reduced optimality and neurodevelopment at 1 month and 3 years of age: Results from the Japan Environment and Children’s Study (JECS)PLOS ONE

Dear Dr. Yasumitsu-Lovell,

Thank you for submitting your manuscript to PLOS ONE. After careful consideration, we feel that it has merit but does not fully meet PLOS ONE’s publication criteria as it currently stands. Therefore, we invite you to submit a revised version of the manuscript that addresses the points raised during the review process.

Please address in particular the limitations raised by the reviewers, in what regards te importance of prematurity as a factor, and those linked to parental assessment of neurodevelopment. 

We look forward to receiving your revised manuscript.

Kind regards,

Umberto Simeoni

Academic Editor

PLOS ONE

Journal Requirements:

“The Japan Environment and Children’s Study (JECS) is funded by the Ministry of the Environment, Japan.”

“No”

6. One of the noted authors is a group or consortium Japan Environment and Children’s Study Group. In addition to naming the author group, please list the individual authors and affiliations within this group in the acknowledgments section of your manuscript. Please also indicate clearly a lead author for this group along with a contact email address.

Reviewers' comments:

Reviewer's Responses to Questions

**Comments to the Author**

1. Is the manuscript technically sound, and do the data support the conclusions?

Reviewer #1: Yes

Reviewer #2: Yes

2. Has the statistical analysis been performed appropriately and rigorously? 

Reviewer #1: Yes

Reviewer #2: Yes

3. Have the authors made all data underlying the findings in their manuscript fully available?

Reviewer #1: No

Reviewer #2: No

4. Is the manuscript presented in an intelligible fashion and written in standard English?

Reviewer #1: Yes

Reviewer #2: Yes

5. Review Comments to the Author

Reviewer #1: This study using Japan's National Children's Study with over 70,000 infants and known prenatal exposures aims to understand the association between pre and peri natal exposures, parental concern for infants and developmental outcomes at 3 years.

25 perinatal states were chosen and the optimal state within each was defined. Each of the 25 items were scored 0 or 1 and appear to be weighted evenly in the analysis.

The one month parent questionnaire was based on ESSENCE shown to correlate with individuals with NDD. Score was analyzed as 0-6 or <2 >= 2

After a very detailed description of the 1 month scoring, the 3 year outcome description is much less informative. NDD were based only on parental report.

The two major concerns I have regarding this manuscript are:

1. the NDD was parental report. ( this can be better described as a weakness in the manuscript)

2. The group was not adjusted for prematurity which is the driving factor in populations at risk for NDD. This weakness needs to be addressed by the authors in order to reach the conclusions that are reached.

Reviewer #2: The strengths of this study include the large sample size and rigorous statistical analysis.

However, several weaknesses also exist. Parent reported outcomes are notoriously inaccurate. The time points of 1 month and 3 years are so far apart in time that many other environmental factors likely have an impact on outcomes. More of the interim data need to be included.

6. PLOS authors have the option to publish the peer review history of their article (what does this mean?). If published, this will include your full peer review and any attached files.

Reviewer #1: **Yes: **An L Anderson Berry, MD, PhD

Reviewer #2: No

---

## [Author Response · Author response to Decision Letter 0]

4 Dec 2022

(Please be informed that the following response is also written in our rebuttal letter.)

Thank you very much to both reviewers for your very insightful and helpful comments. We believe that the following three points need to be improved in the manuscript: 1) limitations regarding parent-reported NDD diagnosis; 2) the time point between 1 month and 3 years of age; and 3) prematurity. Let us respond to each comment as below. 

1) Parent-reported NDD diagnosis: We fully agree with your point, and have elaborated on the third limitation with references as follows: “Third, the information regarding NDD diagnoses was based on the parental questionnaire and was not obtained from a medical database. The reported NDDs are most likely accurate and unlikely to be overreported, because the question regarding the child’s medical history specifically asked about “diagnoses by medical doctors.” However, one cannot exclude a possibility of underreporting due to the stigma and secrecy coping followed by any NDD diagnosis at this early stage of life. (LL418-422)

2) The time point between 1 month and 3 years of age: We also agree with this point as we could not cover all the possible (genetic and environmental) risk factors in this study not only during pre-/perinatal periods but also after birth till three years of age. Therefore, we added the fourth limitation as follows: “Fourth, even though the pre-/perinatal periods are critical windows of neurodevelopment, not all the risk factors were incorporated in our analysis, including those during the first years of child life prior to the NDD diagnosis.” (LL422-427)

3) Prematurity: Thank you very much for pointing this important factor out. As we believe both pre- and post-term are risk, we included 11. Gestational age in the optimality scale, and defined that 37-40 weeks as optimal (Table 1), meaning that both pre- and post-term birth were included in one group as “non-optimal group”. Table 3 shows 10,965 participants were in the non- optimal group regarding gestational age. 

Following reviewer #2’s comment, instead of stating about prematurity in the limitations, we re-categorized the participants into three (pre-term, full-term, and post-term), conducted the same analysis to calculate aRR, and added the following sentences in the manuscript (LL302-314): 

“Similar to the definition of non-optimal maternal age (<20 or >35 years old), non-optimal gestational age at birth included both pre-term (<37 weeks, n = 3,448, 4.9% of the whole participants) and post-term (≥41 weeks n = 6,577, 9.2%). To investigate possible different risk of pre- and post-term birth, gestational age was recategorized into three groups: <37 weeks (pre-term), ≥37to <41 weeks (term), and ≥41 weeks (post-term). The proportion of children with NDDs in each of the three groups were 2.1%, 1.0%, and 1.3% respectively. In the bivariate analysis, crude RRs for the pre- and post-term groups for 3-year NDDs were 2.20 (95%CI 1.73-2.79, p<0.001), and 1.30 (95%CI 1.03 1.64, p<0.05) respectively. However, neither aRRs of the pre- and post-term groups in the final analysis showed statistically significant association (pre-term: aRR 0.81, 95%CI 0.56-1.17, p=0.259; post-term: aRR 1.21, 95%CI 0.95-1.56, p=0.113)”. 

This change on the sub-analysis was also reflected in Statistical Analysis of the method section (LL202-206).

From the mean VIF = 1.14, we concluded that no multicollinearity between the new categorized gestational age and each of the other pre-/perinatal optimal factors was confirmed in the final analysis. No interaction between the recategorized gestational age and other factors was found as well.

Recognizing the wide range among pre-term birth (22 weeks to <37 weeks) and the fact that every day in uterus counts for premature babies, we further divided the pre-term group into three sub-groups (“extremely premature”: <28 weeks, n = 61; “very premature”: ≥28 & <32, n = 252; “moderately premature”: ≥32 & <37, n = 3,175), and found that the same tendency - the more premature they were, the higher prevalence of NDDs (and crude RR) in each group. Post-term was also shown as a risk. The crude RRs of the five groups are as follows: extremely premature 3.40 (95%CI 0.87-13.32, p=0.079); very premature 3.30 (95%CI 1.66-6.54, p=0.001); moderately premature 2.10 (95%CI 1.62 – 2.70, p<0.001); and post-term 1.30 (95%CI 1.02 – 1.63, p=0.028). As the prevalence of the NDDs was just over 1% among the whole sample at this age, and the subcategorization made each group’s number rather small, we have decided not to include the results in this manuscript while acknowledging that the gestational age at birth is one of the most important factors.

Please do not hesitate to contact me should you have any further comments/questions.

Sincerely yours, 

Kahoko Yasumitsu-Lovell

---

## [Decision Letter · Decision Letter 1]

26 Dec 2022

Pre-/perinatal reduced optimality and neurodevelopment at 1 month and 3 years of age: Results from the Japan Environment and Children’s Study (JECS)

PONE-D-22-26962R1

Dear Dr. Yasumitsu-Lovell,

We’re pleased to inform you that your manuscript has been judged scientifically suitable for publication and will be formally accepted for publication once it meets all outstanding technical requirements.

Kind regards,

Umberto Simeoni

Academic Editor

PLOS ONE

Additional Editor Comments (optional):

Reviewers' comments:

Reviewer's Responses to Questions

**Comments to the Author**

1. If the authors have adequately addressed your comments raised in a previous round of review and you feel that this manuscript is now acceptable for publication, you may indicate that here to bypass the “Comments to the Author” section, enter your conflict of interest statement in the “Confidential to Editor” section, and submit your "Accept" recommendation.

Reviewer #1: All comments have been addressed

Reviewer #2: (No Response)

2. Is the manuscript technically sound, and do the data support the conclusions?

Reviewer #1: Yes

Reviewer #2: Yes

3. Has the statistical analysis been performed appropriately and rigorously? 

Reviewer #1: Yes

Reviewer #2: Yes

4. Have the authors made all data underlying the findings in their manuscript fully available?

Reviewer #1: Yes

Reviewer #2: Yes

5. Is the manuscript presented in an intelligible fashion and written in standard English?

Reviewer #1: Yes

Reviewer #2: Yes

6. Review Comments to the Author

Reviewer #1: (No Response)

Reviewer #2: Although the authors did address the previous comments, their efforts did not make the paper more acceptable. I say this primarily due to the gap in between their time points. Too many factors play a role in ND outcomes in 3 years. The parent reported NDD outcome remains very weak in terms of accuracy.

7. PLOS authors have the option to publish the peer review history of their article (what does this mean?). If published, this will include your full peer review and any attached files.

Reviewer #1: **Yes: **Ann Anderson Berry, MD, PhD

Reviewer #2: No

---

## [Editor Report · Acceptance letter]

28 Dec 2022

PONE-D-22-26962R1 

Pre-/perinatal reduced optimality and neurodevelopment at 1 month and 3 years of age: Results from the Japan Environment and Children’s Study (JECS) 

Dear Dr. Yasumitsu-Lovell:

I'm pleased to inform you that your manuscript has been deemed suitable for publication in PLOS ONE. Congratulations! Your manuscript is now with our production department. 

Kind regards, 

on behalf of

Prof. Umberto Simeoni 

Academic Editor

PLOS ONE